# Precious Gene: The Application of RET-Altered Inhibitors

**DOI:** 10.3390/molecules27248839

**Published:** 2022-12-13

**Authors:** Qitao Gou, Xiaochuan Gan, Longhao Li, Qiheng Gou, Tao Zhang

**Affiliations:** 1Department of Oncology, The First Affiliated Hospital of Chongqing Medical University, 1 Youyi Road, Yuzhong, Chongqing 400016, China; 2Department of Radiotherapy, Cancer Center, West China Hospital, Sichuan University, Chengdu 610041, China

**Keywords:** molecular target therapy, small molecule inhibitor, tyrosine kinase inhibitor, RET-selective inhibitor, selpercatinib, pralsetinib, TPX-0046, zetletinib

## Abstract

The well-known proto-oncogene rearrangement during transfection (RET), also known as ret proto-oncogene Homo sapiens (human), is a rare gene that is involved in the physiological development of some organ systems and can activate various cancers, such as non-small cell lung cancer, thyroid cancer, and papillary thyroid cancer. In the past few years, cancers with RET alterations have been treated with multikinase inhibitors (MKIs). However, because of off-target effects, these MKIs have developed drug resistance and some unacceptable adverse effects. Therefore, these MKIs are limited in their clinical application. Thus, the novel highly potent and RET-specific inhibitors selpercatinib and pralsetinib have been accelerated for approval by the Food and Drug Administration (FDA), and clinical trials of TPX-0046 and zetletinib are underway. It is well tolerated and a potential therapeutic for RET-altered cancers. Thus, we will focus on current state-of-the-art therapeutics with these novel RET inhibitors and show their efficacy and safety in therapy.

## 1. Introduction

In the past few years, research on targeted oncogene drivers has become increasingly in-depth. The first generation of kinase inhibitors, such as breakpoint cluster region-Abelson murine leukemia fusions (BCR-ABL), epidermal growth factor receptor mutations (EGFR), and anaplastic lymphoma kinase rearrangements (ALK), confirmed that they have great therapeutic potential in chronic myeloid leukemia and kinase-driven non-small cell lung cancer (NSCLC) [1,2,3]. These targeted kinase inhibitors have successfully used a new therapeutic mode to benefit specific patients. However, drug resistance in therapy is a major clinical challenge, which encourages people to strive to develop more effective and selective next-generation kinase inhibitors [4,5,6].

Rearrangement during transfection (RET) is a widely known proto-oncogene that drives various cancers. RET plays a critical role in the fatal development of the hematopoietic, nervous, gastrointestinal, and genitourinary systems [7,8,9,10]. Activating RET aberration is a potent carcinogenic driver and drives tumorigenesis, such as lung cancer and thyroid cancer [11,12]. To date, two major mechanisms of RET kinase activation have been described: RET point mutations and RET gene rearrangement or fusion. RET point mutations are most commonly found in sporadic medullary thyroid cancer (MTC) and as germline mutations in multiple endocrine neoplasia syndrome (MEN). Approximately 25–40% of sporadic MTCs have to activate RET mutations, and approximately all cases of familial MTC also have to activate RET mutations [13,14,15]. RET gene rearrangement or fusion occurs in 1–2% of NSCLC and 10–20% of papillary thyroid carcinoma (PTC) [13,16].

In the past few years, the treatment of RET-altered cancer has mainly focused on MKIs. Although these MKIs were originally designed to target other kinases (EGFR, ALK, cellular-mesenchymal to epithelial transition factor (MET), vascular endothelial growth factor receptor 2 (VEGFR-2), etc.), these agents have been repurposed to treat patients with RET alterations, such as cabozantinib, vandetanib, lenvatinib, sorafenib, and sunitinib. However, each agent has produced limited disease control and lower response rates and might easily cause high-grade adverse events. Therefore, only cabozantinib and vandetanib are approved by the FDA to treat locally advanced or metastatic MTC or NSCLC patients with RET and allow researchers to find more effective agents, such as selective inhibitors [17,18,19]. In 2020, the FDA approved two RET-selective inhibitors (selpercatinib and pralsetinib) to treat patients with RET-driven MTC and NSCLC [20,21,22,23,24]. To date, these selective inhibitors have given patients more and better choices in treatment. In this article, we review recent targeted drugs, including RET-altered cancer, and raise issues that require further study.

## 2. Mechanism and Biology of RET

In 1985, Takahashi et al. accidentally found a novel transforming gene detected by transfection of NIH-3T3s cells with human lymphoma DNA, and this transforming oncogene was named RET [25]. By 1989, RET was mapped to the long arm of 10q11.2 chromosome 10. It is composed of an extracellular domain, a transmembrane domain, and an intracellular domain [26]. Four cadherin-like domains (CLD1–4), a calcium binding site, and a cysteine-rich domain composed the extracellular domain. At the end of the C-terminal domain of RET, there are two major isoforms in vivo: one has nine amino acids (RET 9), another has 51 amino acids (RET 51), and RET 51 has stronger tumorigenic activity than RET 9 [27]. The ligand of the RET tyrosine receptor is a member of the glial cell line-derived neurotrophic factor family ligands (GFLs). There are four known ligands: glial cell line-derived neurotrophic factor (GDNF), neurturin (NRTN), artemin (ARTN), and persephin (PSPN) [28]. These GFLs bind to the GDNF family receptor alpha (GFRα). Then, the dimerization of RET is induced by the GFL-GFRα complex, which can bind to the intracellular tyrosine kinase domain [29]. After that, dimerization of RET activates a variety of cellular signal cascades, such as the RAS/MAPK, PI3K/AKT, JAK/STAT, PKA, and PKC pathways [30,31]. These pathways are the main causes of cell survival, proliferation, migration, and differentiation [32,33]. (Figure 1)

Ret fusion or rearrangements occur when RET 3′ sequences that encode the intracellular tyrosine kinase domain juxtapose with 5′ sequences from another gene encoding protein dimerization domains when translocation or chromosomal inversion occurs [14]. RET fusion or rearrangement accounts for 5–10% of PTCs. The two most prevalent fusions or rearrangements are in the coil–coil domain containing 6 (CCDC6)-RET and nuclear receptor coactivator 4 (NCOA4)-RET, and these aberrations account for more than 90% of rearrangements [34,35,36,37]. Of interest, radiation-exposed populations are more prevalent and carry RET fusions/rearrangement. For example, RET fusion or rearrangement was more likely detected in PTC and survived in Chernobyl-contaminated areas and the Hiroshima or Nagasaki atomic bomb explosions [38,39,40]. In addition, approximately 2% of patients with NSCLC carry RET fusions. Kinesin family member 5B (KIF5B)-RET (70–90%) is the most common RET fusion in lung cancer, and CCDC6-RET (10–25%) also has a high proportion. Until now, more than 15 RET fusions have been reported in NSCLC [16,41,42]. Of interest, RET rearrangement can coexist with activated EGFR mutations in EGFR-mutated NSCLC patients who have progressed on first- or second-generation EGFR- tyrosine kinase inhibitors (TKIs). These patients showed drug resistance to sunitinib, lenvatinib, vandetanib, and sorafenib [43]. Moreover, in other solid tumors, RET fusion and rearrangement can also be discovered in colorectal carcinomas, salivary gland adenocarcinomas, etc., although these diseases account for a small proportion [44,45].

When the RET point mutation occurs, these abnormal sites result in aberrant RET kinase transcription. In mutated RET kinase, their replacement with other amino acids leads to the formation of aberrant intermolecular disulfide bonds. This abnormal structure causes ligand-independent RET kinase dimerization and kinase activation [46]. A RET point mutation is more common in thyroid cancer than NSCLC and is more likely to occur in MTC, especially multiple endocrine neoplasia type 2 (MEN2) [45]. According to the clinical characteristics, two subtypes can be identified: MEN2A and MEN2B. Familial medullary thyroid carcinoma (FMTC) was previously considered a third MEN2 subtype but is now considered part of the MEN2A subtype [47]. The most frequent RET mutation in MEN2A was at codon 634 (30%). In addition, RET-M918T is the most common and most aggressive mutation out of more than 60 activating RET mutations discovered. Approximately 95% of MEN2B and 40% of sporadic MTC are associated with M918T [12,48]. RET point mutations also exist in a variety of cancers, such as breast cancer (C634R), colorectal adenocarcinoma (V840 M), and gastrointestinal stromal tumor (V840 M) [49].

## 3. RET-Targeted Therapies

### 3.1. Multikinase Inhibitors (MKIs)

#### 3.1.1. MKIs in Lung Cancer

MKIs that target the RET pathway are similar to other MKIs in the structure of the kinase domain. In the early stage, studies on targeted therapy for tumors with positive RET fusion mainly focused on RET-TKI activity, such as vandetanib, cabozantinib, and lenvatinib. An open-label multicenter phase II clinical trial (LURET, UMIN000010095) included 17 eligible patients. The overall response rate (ORR) of nine patients treated with vandetanib was 53% (95% CI 28–77), and the median progression-free survival (PFS) was 4.7 months (95% CI 2.8–8.5). The subgroup efficacy analysis indicated that five out of six patients (83%, 95% CI 36–100) with CCDC6-RET achieved an objective response, and two out of ten patients (20%, 95% CI 3–56) with KIF5B-RET achieved an objective response. Most of the drug-related adverse events were grades 1–2, and the most common treatment-related grade 3–4 adverse event was hypertension (84%) [50]. In addition, a phase II, single-arm clinical trial (NCT01639508) of 26 patients with metastatic or unresectable RET fusion adenocarcinoma was performed. This study demonstrated that cabozantinib is active in patients with advanced RET-rearranged lung cancers, with an overall response rate of 28%. Nineteen patients (73%) required dose reduction because of drug-related adverse events. The most common grade 3–4 adverse event was asymptomatic lipase elevation (15%) [51]. A global multicenter clinical trial (GLORY) retrospectively collected the efficacy of multiple MKIs used by patients who were diagnosed with NSCLC with RET rearrangements. Of the TKI-naïve patients, the ORR for cabozantinib (*n* = 19) was 37%, vandetanib (*n* = 11) was 18%, and sunitinib (*n* = 9) was 22%. Furthermore, one patient achieved a partial response (PR) with lenvantinib (*n* = 2), and one patient achieved a complete response (CR) with nintedanib (*n* = 2). No response was observed in the sorafenib group (*n* = 2), alectinib group (*n* = 2), lenvartinib group (*n* = 2), ponatinib group (*n* = 2), or regorafenib group (*n* = 1). The median PFS was 2.3 months (95% Cl 1.6–5.0), and the median overall survival (OS) was 6.8 months (95% Cl 3.9–14.3) [49]. Agreafenib (RXDX-105) is a novel MKI that can potently inhibit RET mutations, various RET-associated fusions, and select mutant proteins (e.g., M918T). An important phase I/Ib clinical study showed that sorafenib has favorable efficacy in RET-altered NSCLC patients. The ORR was 19% (95% Cl 8–38), and no one was assessed to have a complete response. Of interest, patients with the KIF5B-RET fusion gene have a worse ORR than those with the non-KIF5B-RET fusion gene (0% versus 67%) [52].

#### 3.1.2. MKIs in Thyroid Cancer

In recent decades, several drugs or combinations of drugs have been approved by the FDA for thyroid cancer, and many other drugs have also been researched for thyroid cancer. Agents targeting the RET receptor, such as vandetanib, cabozantinib, lenvatinib, and sorafenib, can achieve clinical survival benefits. The famous Phase III clinical trial named Zeta (NCT00410761) demonstrated that vandetanib can bring survival benefits to patients with locally advanced/unresectable or metastatic MTC. It included 331 patients who were assigned to receive vandetanib or a placebo. The study indicated that patients who received vandetanib had a significantly longer median PFS than patients who received the placebo (30.5 m vs. 19.3 m) [53]. In addition, a phase III trial (EXAM; NCT00704730) also demonstrated that patients receiving cabozantinib had a better objective response rate (ORR) than the placebo (11.2 m vs. 4.0 m) in progressive medullary thyroid cancer, and the response rate was 28% for the cabozantinib group [51]. Vandetanib and cabozantinib have been approved for first-line treatment in MTC by the ZETA trial and the EXAM trial. In a real-world multicenter cohort in Germany, vandetatinib and cabozantinib also showed significant efficacy in advanced MTC patients. An mPFS of up to 47 months for vandetanib and up to 4 months for cabozantinib was indicated, and the median OS was also good for vandetanib (53 m) and cabozantinib (24 m) [54].

Meanwhile, some studies also provide evidence that MKIs can give patients better survival benefits. In a phase III clinical study (DECISION; NCT00984282), the patients with DTC without kinase inhibitors received sorafenib, and the median PFS in the sorafenib arm was higher than that in the placebo arm (10.8 m vs. 5.8 m). However, the median OS in the sorafenib group was similar to that in the placebo group [55]. Another clinical trial was assigned to lenvatinib and placebo groups in radioiodine-refractory thyroid cancer (SELECT, NCT01321554). There were significant PFS (18.3 m vs. 3.6 m) and ORR (64.8% vs. 1.5%) in the lenvatinib group, but no differences in median OS were reported [56].

#### 3.1.3. MKI in Other Solid Tumors

NCOA4-RET is one of the RET-altered fusions in colorectal cancer (CRC) and occurs with only a 0.2% frequency [44]. A study demonstrated that vandetanib has efficacy in MTT proliferation assays, and other MKIs did not suppress cell viability. This means that vandetanib might be a useful TKI for CRC patients with the NCOA4-RET fusion in carbozantinib, sorafenib, vandetanib, and PD0332991 [45]. The RET gene and RET mutation or fusion also occur in breast cancer. However, MKI has been shown to effectively block RET signaling in in vitro and in vivo studies. Many MKIs, such as imatinib, sorafenib, sunitinib, and vandetanib, did not reach the expected efficacy in clinical trials [57]. These studies demonstrated that MKI does not have favorable results in other RET-associated solid tumors.

#### 3.1.4. Disadvantages of MKIs

MKIs not only target RET but also target other kinases, such as EGFR, VEGFR2, human epidermal growth factor receptor-2 (HER-2), MET, etc. [58]. In particular, the domain of VEGFR-2 kinase was highly similar to RET. Therefore, several MKIs (vandetanib, cabozantinib, Lenvatinib, and sorafenib) can show therapeutic potential for cancer patients [51,53,55,56]. Unfortunately, off-target effects can also limit the efficacy of these MKIs. Moreover, the upstream partner genes KIF5B and RET can fuse and cause intrinsic resistance. Therefore, these MKIs have developed resistance and caused the treatment discontinuation rate of drugs. In a phase I/Ib trial, treatment with sorafenib had less efficacy in NSCLC patients with KIF5B-RET genes than in those without the KIF5B-RET fusion gene [52]. In addition, these MKIs have drug-related toxicity and might increase the dose-reduction rate and treatment-discontinuation rate, influencing their clinical application [10,42,59]. Because of limited treatment efficacy and a high probability of adverse events, only vandetanib and cabozantinib are recommended by the NCCN guidelines to treat metastatic RET fusion-positive NSCLC. However, both drugs failed to achieve their goals in the exploration of NSCLC, and off-target effects also limited clinical efficacy [60]. On the other hand, these clinical trials demonstrate that patients can acquire survival benefits from the RET gene. This finding promotes the conversion from MKIs to RET-selective inhibitors.

### 3.2. Selective RET Inhibitors

MKIs have lots of disadvantages, such as adverse events and drug resistance. These disadvantages can significantly affect therapeutic efficacy and limit clinical application. Thus, researchers have developed a new generation of RET inhibitors. Recently, two new generation, highly RET-specific, gatekeeper mutant-effective TKIs (pralsetinib and selpercatinib) showed a satisfactory survival benefit in clinical trials (ARROW and LIBRETTO-001). Therefore, they are approved by the FDA for the treatment of RET-altered NSCLC and thyroid cancers. TPX-0046, BOS-172738, and RXDX-105 are novel, orally administered, RET-specific TKIs. These agents can inhibit the wild-type RET gene and variant RET mutations, and related studies are ongoing. (Figure 2)

#### 3.2.1. Selpercatinib (LOXO-292)

Selpercatinib is a novel, ATP-competitive, highly selective small-molecule inhibitor of RET kinase. It was approved for the treatment of NSCLC and thyroid cancer patients with RET alterations by the FDA on May 8, 2020. Wild-type and variant RET mutations and some kinases, such as VEGFR1 and VEGFR3, can be suppressed by selpercatinib. It can also selectively inhibit a high level of KIF5B-RET or CCDC6-RET fusion gene activity. In addition, a study showed that selpercatinib was effective in V804 L and V804 M gatekeeper mutations and was found to be 60–1300-fold more effective than MKIs in engineered cell lines that have KIF5B-RET V804 L/M gatekeeper mutations [61]. It also shows activity against VEGFR1 and VEGFR3 and lower activity against VEGFR2 [62]. In addition, a report indicated that selpercatinib can also inhibit L730 V/I RET roof mutations and is more effective than pralsetinib [63]. In preclinical studies. Selpercatinib caused significant tumor regression compared with cabozantinib in mouse models. Compared to ponatinib, selpercatinib also caused significantly prolonged survival in mice that had intracranial CCDC6-RET fusion positivity. This means that selpercatinib has the ability to penetrate the blood-brain barrier and eliminate tumor cells in brain tissue [64].

The pivotal clinical trial LIBRETTO-001 (NCT03157128) [21]. Both treatment groups of selpercatinib exhibited a good survival benefit for NSCLC patients. By 2022, 247 NSCLC patients had previously been treated with at least platinum-based chemotherapy. The objective response rate (ORR) was 61% (95% confidence interval [CI], 55–67); 7% (*n* = 18) of patients had a complete response (CR), and 49% responded. The median duration of response (DOR) was 28.6 months (95% CI, 20.4-NE). The median progression-free survival (PFS) was 24.9 months (95% Cl 19.3-NE), and 38% of patients who were still alive were progression free. In addition, the 1-year PFS was 70.5%, and the 2-year PFS was 51.4%. In addition, the 2-year OS rate was 69%. In addition, in 69 newly diagnosed NSCLC patients in another group, the ORR was 84% (95% CI, 73–92), four (6%) patients achieved CR, and 54 (78%) patients achieved PR. The mDOR was 20.2 m (95% Cl, 13.0-NE), and the mPFS was 22 m (95% Cl, 13.8-NE). The median OS was not estimable, and the 2-year OS rate was 69%. In this clinical trial, patients who were diagnosed with NSCLC obtained meaningful clinical benefits, even though most patients had received previous therapy. The most common adverse events of grade 3 were abnormal ALT/AST levels (20.2%), hypertension (13.2%), diarrhea (5.0%), and prolonged electrocardiogram QT (4.8%). In addition, approximately 8% of patients discontinued selpercatinib due to serious side effects, and one patient died due to acute respiratory failure because of treatment with selpercatinib [65]. However, this was different from a retrospective study that included 50 patients with RET fusion-positive advanced NSCLC treated with selpercatinib. The most common grade ≥3 side effects were increased liver enzyme levels (10%), prolonged QTc time (4%), abdominal pain (4%), hypertension (4%), and fatigue (4%). Notably, no agents were discontinued because of drug-related adverse events [62]. In the real-world study, the most common adverse events were slightly different from those in the LIBRETTO-001 trial, and the real-world data on side effects were better than those of the clinical trial. We think that this retrospective study is a small sample study, which might have caused some bias. In addition, practitioners might use some adjuvant treatment that decreases the side effect rate. From this research result, the effectiveness of selpercatinib is better than expected, and we expect a real-world study of a large sample to prove this conjecture. Regarding side effects, practitioners should focus on abnormal liver enzymes when using selpercatinib [62]. Brain metastasis is prevalent in lung cancer patients, and selpercatinib has significant CNS penetration in RET fusion-positive lung cancer. Therefore, some research on brain metastasis has been conducted. Thirty-eight of 105 patients were diagnosed with brain metastases, and 11 out of 38 patients were deemed to have measurable lesions. The ORR of brain metastasis patients was 91% (*n* = 10, 95% CI 59–100). Notably, three patients (27%) had CR, and seven patients (64%) had PR. The median central nervous system (CNS) DOR was 10.1 months (95% Cl 6.7-NE) [66].

RET mutations account for a high proportion of mutations in thyroid patients. The LIBRETTO-001 trial also demonstrated that selpercatinib had significant and long-lasting activity in MTC patients. In 88 patients who had not previously received targeted therapy, the ORR was 73% (95% CI 62–82). The 1-year PFS was 92% (95% Cl 82–97). In the 55 patients who had previously received vandetanib and cabozantinib, the ORR was 69% (95% CI 55–81) and the 1-year PFS was 82% (95% CI 69–90). In addition, 19 patients with RET fusion-positive thyroid cancer had previously been treated with therapy. The ORR was 79% (95% Cl 54–94), and the 1-year PFS rate was 64% (95% Cl 37–82) [67].

Selpercatinib also showed a significant benefit for RET fusion-positive solid tumors. Of 41 patients with efficacy-evaluable tumors, the ORR was 43.9% (95% CI 28.5–60.3). Preliminary studies have shown that selpercatinib has benefits, including 2 CR and 16 PR. The 1-year PFS was 53.1% (95% Cl 34.1–68.8), and the 2-year PFS was 32.1% (95% Cl 14.0–51.7). Notably, 59% (*n* = 24) of patients were diagnosed with gastrointestinal malignancies [68]. This means that selpercatinib has therapeutic potential for gastrointestinal tumors with RET. Meanwhile, selpercatinib also has excellent therapeutic potential in pediatric cancer patients. A preliminary study included five pediatric patients who were diagnosed with two thyroid cancers and three soft-tissue sarcomas. It has been shown that treatment with selpercatinib can achieve a four-fifths partial response, and one achieved stable disease [69]. In addition, another study also demonstrated that using selpercatinib can have excellent therapeutic efficacy in pediatric patients, which included four patients (infantile myofibroma/hemangiopericytoma, *n* = 1; congenital mesoblastic nephroma/infantile fibrosarcoma, *n* = 1; lipofibromatosis, *n* = 1; papillary thyroid cancer, *n* = 1), and two patients had a partial response [70]. Furthermore, several clinical trials are also ongoing, such as LIBRETTO-431 (NCT04194944), LIBRETTO-121 (NCT03899792), and LIBRETTO-321 (NCT04280081). (Table 1)

#### 3.2.2. Pralsetinib (BLU-667)

Pralsetinib is also an orally available, highly selective RET inhibitor, and it demonstrated antitumor activity in patients with RET-altered NSCLC, thyroid cancer, and another solid tumor. The agent received approval in September 2020 for the treatment of NSCLC and thyroid cancer [23]. It was found to be similar to selpercatinib, which has greater potency against RET fusions and mutants, such as KIF5B-RET and CCDC6-RET. Pralsetinib also possessed potency against clinically relevant mutations found in the gatekeeper region, including V804 L, V804 M, and V804E. It can suppress proliferation, RET-altered fusion, and mutation *in vitro*. Compared to the first-generation kinase inhibitors, pralsetinib was 100-fold more selective for most MKIs and was demonstrated to inhibit RET autophosphorylation in Ba/F3 cells harboring the KIF5B-RET fusion. Pralsetinib is more than 10 times more potent than vandetanib, cabozantinib, and RXDX-105. Researchers have demonstrated that pralsetinib is approximately 88 times more selective for RET than VEGFR2. MKIs have significant VEGFR2 activity. Although VEGFR2 is antiangiogenic, inhibiting VEGFR2 can suppress some solid tumors. However, over-inhibition of VEGF2 has cardiotoxicity, which limits the therapeutic benefits of nonselective MKI for RET-driven diseases [71]. In addition, MKIs have significant other drug-related toxicity and are more likely to cause serious, unacceptable toxicity that limits the amount and duration of therapy [72]. Similar to selpercatinib, pralsetinib aims to overcome the limitations of this treatment and improve the therapeutic effect by targeting multiple clinically relevant RET mutations.

The pivotal clinical trial named Arrow is a global phase I/II clinical trial of pralsetinib (NCT03037385) in RET-related cancers. This trial enrolled 233 patients who were diagnosed with RET fusion-positive NSCLC, 87 patients with previous platinum-based chemotherapy, and 27 patients who were newly diagnosed. The ORR was 61% (95%Cl 50–71) in previous platinum-based chemotherapy. Among the 27 patients in the no previous systemic treatment group, the ORR was 70% (95% Cl 50–86) [20]. In an updated clinical trial (AcceleRET; NCT04222972), 281 patients with RET-fusion NSCLC were included. The ORR was 72% (95%Cl 60–82) versus 59% (95%Cl 50–67) between treatment-naïve patients and those who had received prior chemotherapy. Of interest, tumor shrinkage was observed in 97% of patients with prior chemotherapy and in all treatment-naïve patients. Furthermore, pralsetinib is as good as selpercatinib in the treatment of brain metastasis and pharmacology, and both selective RET inhibitors can cross the blood-brain barrier and shrink intracranial mass. Notably, the intracranial response rate was 70% (95% Cl 35–93) [73].

The ARROW trial also enrolled 122 patients with RET-mutant medullary thyroid carcinoma and 20 with RET fusion-positive thyroid cancer. The data demonstrate that treatment with pralsetinib achieved an ORR of 71% (95% Cl 48–89) in 21 MTC patients who were treatment-naïve RET-mutant MTC. Additionally, treatment with pralsetinib resulted in 60% (95%Cl 46–73) in patients who had previously received targeted therapy and 60% (95%Cl 46–73) with ORR in patients with RET fusion-positive thyroid cancer [74]. This clinical trial promoted the approval of RET-altered thyroid cancer by the FDA.

Pralsetinib was used for the treatment of solid tumors with RET alterations, except thyroid cancer and NSCLC, in 29 patients. The ORR was 57% (95%Cl 35–77) among these patients. Of interest, 91% of patients achieved target tumor shrinkage, including three patients (13%) who achieved CR and ten patients (43%) who achieved PR. The DOR, PFS, and OS were 12 months, 7 months, and 14 months, respectively [75]. These data validated the potential of pralsetinib as a well-tolerated treatment option with rapid, robust, and durable antitumor activity in patients with diverse RET fusion-positive solid tumors. In addition, several relevant studies are ongoing, including NCT04760288, NCT04204928, NCT04697446, and NCT04302025. (Table 1)

#### 3.2.3. TPX-0046

In 2020, a case report indicated that two patients were treated with selpercatinib for disease progression. The researcher analyzed the reason that the acquired RET G810R/S/C solvent front mutations caused drug resistance. The autopsy indicated that G810 residue convergent evolution causes a mechanism of resistance [76]. Therefore, the most important problem to be solved by RET-TKIs is drug resistance. Researchers will not only understand the mechanism of kinase inhibitor resistance but also develop next-generation drugs to overcome drug resistance. TPX-0046 is a novel, orally administered, dual RET/SRC inhibitor that was developed by Turning Point Therapeutics Inc. and has strong potency on RET G810 solvent-front RET mutants. It is a unique agent that possesses potent in vitro and in vivo activity against a diverse range of RET alterations, such as solvent-front mutation resistance [77]. TPX-0046 has a small and rigid macrocyclic structure that is different from that of other RET-selective inhibitors. This macrocyclic can generate a compact type I inhibitor that binds to the ATP-binding site and maintains antitumor activity without drug resistance [72]. Moreover, antitumor drugs have cardiovascular toxicity because they can inhibit VEGFR kinase. Many patients often experience cardiovascular events in the days or years after chemotherapy or targeted therapy, which also increases the difficulty of treatment for people with cardiovascular disease. Of interest, TPX-0046 does not inhibit VEGFR kinase. Therefore, it can avoid cardiovascular toxicities such as hypertension and has better safety [78]. A single-arm, phase I/II trial (Sword-1; NCT04161391) is ongoing to determine the efficacy and safety of TPX-0046 in patients with RET-altered advanced or metastatic solid tumors. Preliminary clinical data were collected from 21 patients in this study, including 10 with NSCLC and 11 with MTC, as of 10 March 2021. Of five RET TKI-naïve patients, PR was achieved in four patients. Of interest, in nine patients previously treated with TKIs, three patients were assessed for tumor regression. The most frequent treatment-emergent adverse event (TEAE) was grade 1 or 2 dizziness, and there were infrequent dose-reduction rates and treatment-discontinuation rates due to TEAEs. These initial preliminary data demonstrated safety and efficacy and encouraged follow-up studies [79].

#### 3.2.4. Zeteletinib (BOS-172738; DS-5010)

Zeteletinib is a novel, orally available, small-molecule RET-selective inhibitor with nanomolar potency against RET and more than 300-fold selectivity against VEGFR2. In addition, it shows favorable potency for wild-type RET and the RET mutation M918T. Furthermore, the V804 L and V804 M gatekeeper mutants can also be inhibited. A phase I study (NCT03780517) included 67 patients with RET-altered advanced solid tumors and demonstrated that the agent has broad antitumor activity. The ORR was 33% (*n* = 18/54). In NSCLC patients, the ORR was 33% (*n* = 10/30). The MTC patients’ ORR was 44% (*n* = 7/16). Notably, one patient was diagnosed with pancreatic cancer with RET fusion and was evaluated for a partial response after treatment with zetletinib. Of interest, zetletinib also shows potency for CNS metastases. One patient had brain metastases, which were decreased by 43% due to zetletinib. In addition, it exhibits significant safety for long-term administration. Most adverse events were classified as grade ≤2 and deemed not related to the agent [80,81,82].

## 4. Future Expectations

The ALK mutation is a rare mutation and is known as the diamond mutation in NSCLC. It has been treated with tremendous success since it was first reported in 2007. ALK-selective inhibitors such as alectinib and lorlatinib give patients significant responses and survival rates [2,83]. Interestingly, RET mutation or fusion is also a rare gene in RET-altered solid tumors, and two clinical trials showed significant success in related patients. MKIs, selpercatinib, and pralsetinib showed favorable ORRs and OS rates in RET-altered cancer, but the current challenge for RET precision medicine should be solved as quickly as possible. Treatment with RET-selective inhibitors led to the development of acquired resistance, although these alterations occurred at a low frequency. A study reported that most of the cases progressing on RET-selective inhibitors might be driven by off-target effects, RET-independent mechanisms of resistance [84]. Researchers should ultimately develop new generations of selective inhibitors to give patients new hope. Therefore, some other new agents have been developed. TAS0953/HM06 is a novel, small-molecule RET-TKI that can inhibit RET-associated NSCLC, papillary and medullary thyroid cancer, and several other solid tumors. It is different from other RET inhibitors in structure and has better blood–brain barrier penetration. A pivotal phase I/II clinical study that includes advanced solid tumors harboring RET fusions or mutations is also underway (NCT04683250). SYHA1815 can inhibit wild-type RET and overcome V804 mutation kinase activity. It has nearly 20-fold selectivity for RET over VEGFR2, and a phase I trial is ongoing in China [85]. LOX-18228 is a selective next-generation RET inhibitor. It can inhibit multiple RET mutations. A preclinical study demonstrated potential therapy with the V840 M mutation and the G810S mutation [86]. Other potent and selective RET inhibitors, such as LOX-19260 and SL-1001, are still in the preclinical stage. The RET gene might be the next diamond site in the future. (Table 1)

## 5. Conclusions

In conclusion, for patients with RET fusion-positive solid tumors, chemotherapy has many limitations. In addition, immunotherapy for patients also has poor effects. Currently, the treatment of RET fusion-positive tumors has entered a new era of targeted treatment. However, because of the off-target effects of MKIs, they easily generate adverse effects and drug resistance. Only a few drugs are approved by the FDA. Thus, MKIs are limited in clinical application. Currently, novel, highly RET-selective inhibitors, such as pralsetinib and selpercatinib, have fundamentally changed the treatment pattern of RET fusion-positive NSCLC patients and thyroid cancer. In the ARROW study and the LIBRETTO-001 study, the use of pralsetinib and selpercatinib resulted in long-lasting antitumor activity and fewer adverse events and showed a significant response rate. These two agents have been recommended by the NCCN guidelines for NSCLC and thyroid cancer with RET fusion. Similar to the ALK gene, known as the diamond gene, the RET alternative might be a new diamond gene in the future. In the future, researchers should focus on solving the problem of drug resistance and side effects. Therefore, researchers and practitioners should also evaluate novel selective RET-TKIs such as TPX-0046 and zeteletinib so that the patient can have more choices and gain better survival benefits.

## Figures and Tables

**Figure 1 molecules-27-08839-f001:**
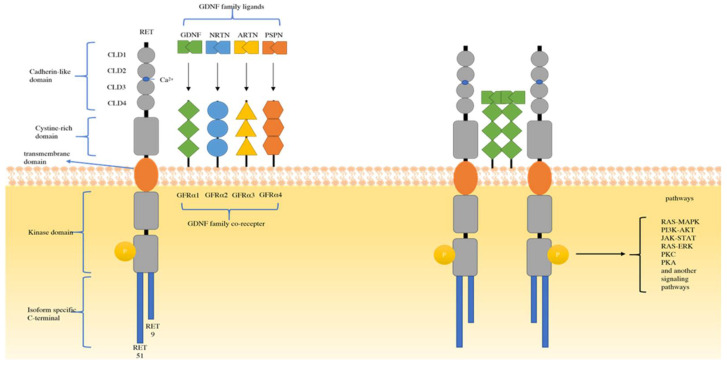
Structure of RET and the major mechanisms of RET activation. RET, rearrangement during transfection; CLD, cadherin-like domains; GFRα, growth factor receptor-alfa; GDNF, glial cell line-derived neurotrophic factor; ARTN, artemin; NRTN, neurturin; PSPN, persephin; Ca^2+^, calcium ion.

**Figure 2 molecules-27-08839-f002:**
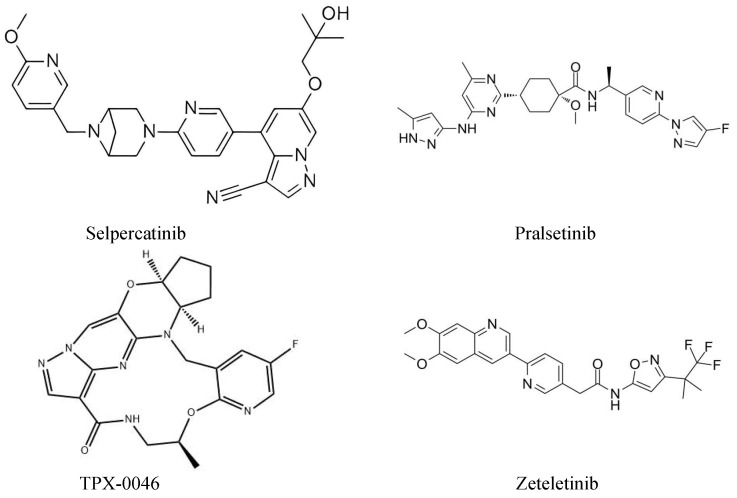
Chemical structure of selpercatinib, pralsetinib, TPX-0046, and zetletinib.

**Table 1 molecules-27-08839-t001:** Clinical trial of RET selective inhibitor for cancer. CNS, central nervous system; MTC, medullary thyroid carcinoma; NSCLC, non-small cell lung cancer; N/A, not applicable; NCT, National Clinical Trials.

Drug	Cancer	Phase	Location	Statue	NCT
Selpercatinib					
	MTC, Solid tumor	II	China	active	NCT04280081(LIBRETTO-321)
	RET-altered advanced solid tumors, lymphomas, and histiocytic disorders in pediatric patients	II	US	recruiting	NCT04320888
	NSCLC	III	global	recruiting	NCT04819100(LIBRETTO-432)
	NSCLC, MTC, colon cancer, advanced solid tumor	I/II	global	recruiting	NCT03157128 (LIBRETTO-001)
	MTC	III	global	recruiting	NCT04211337 (LIBRETTO-531)
	thyroid cancer	II	US	recruiting	NCT04759911
	Advanced or Metastatic RET Fusion-Positive NSCLC	III	global	recruiting	NCT04194944 (LIBRETTO-431)
	NSCLC	II	US	recruiting	NCT05364645
	NSCLC	II	US	active	NCT04268550
	NSCLC, MTC, colon cancer, another solid tumor	N/A	global	available	NCT03906331
	Advanced solid tumor and Primary CNS tumors in pediatric patients	I/II	global	recruiting	NCT03899792 (LIBRETTO-121)
	NSCLC	II	global	recruiting	NCT03944772 (ORCHARD)
	Advanced cancer, solid tumor	II	Finland	recruiting	NCT05159245 (FINPROVE)
	Advanced solid tumor, lymphomas, histiocytic disorders in pediatric patients	II	US	recruiting	NCT03155620
Pralsetinib					
	MTC	III	Spain	Not yet recruiting	NCT04760288(AcceleRET-MTC)
	NSCLC, MTC	N/A	N	approved	NCT04204928
	NSCLC, MTC, another solid tumor	I/II	global	active	NCT03037385 (ARROW)
	NSCLC	III	global	recruiting	NCT04222972 (AcceleRET-Lung)
	NSCLC	N/A	US, France, Switzerland	enrolling	NCT04697446
	NSCLC	III	global	recruiting	NCT05170204
	NSCLC	II	US	recruiting	NCT04302025
	Advanced cancer, solid tumor	II	Finland	recruiting	NCT05159245(FINPROVE)
	Advanced Unresectable or Metastatic Solid Malignancy	II	US	recruiting	NCT04632992 (MyTACTIC)
	Solid tumor	II	global	recruiting	NCT04589845
TPX-0046					
	Advanced solid tumors	I/II	US, South Korea	recruiting	NCT04161391
TAS0953/HM06					
	Advanced solid tumors	I/II	US, Japan	recruiting	NCT04683250
SYHA1815					
	Advanced or metastatic solid tumors	N/A	China	recruiting	NCT05105464

## Data Availability

The data used to support the findings of this research are available from the first author upon request.

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
