# Peer review of "Precious Gene: The Application of RET-Altered Inhibitors"

_molecules, 2022, doi:10.3390/molecules27248839_

Round 1

Reviewer 1 Report

In the submitted review article (Precious gene: the application of RET-altered inhibitors), the authors reviewed the existing literature on current (and expected) state-of-the-art therapeutic RET-specific inhibitors.

Congratulations to the authors on an excellently treated, very actual research/scientific/pharmacology/medicinal topic, as the text is well designed and written and (most importantly) more than useful to everyone interested in this therapeutic type.

I would make minor remarks to make the final text even more informative.

I suggest you add a picture showing the chemical structure of all mentioned (apparently) highly potent and well-tolerated RET-altered inhibitors (e.g., selpercatinib, pralsetinib, TPX-0046, zetletinib). I suggest adding a picture showing the chemical structure of all mentioned (apparently) highly potent RET-altered inhibitors. This information is important for (organic) chemists and biochemists.

Two sentences are inappropriate beginning parts of the text... please correct them. The first is on line 208, and the second starts on line 233.

Please change the title of Figure 1. It shows a schematic representation of the structure of RET (and activation mechanism). The first letter must be capitalised in this figure and in the title of Table 1. Please use Ca2+ (number and plus sign in superscript), not Ca++.

Almost the entire text lacks a space between the last word in the sentence, and the parentheses for the reference cited (lines 31, 34, 37, etc.)

Lines 290 and 291: according to the rest of the text, write n=1, not n = 1 (so no spaces)! Next, move the period/dot to the end of the sentence (lines 76 and 294, the end of the fourth paragraph of section 3.2.2. and section 4).

The name of the drug is incorrectly stated (typing error) in the sentence in the second paragraph of section 3.2.2. (Furthermore, ...).

Check the correctness of quotations 75, 78 and 84 (conference papers)... I believe that these are summaries on only one page.

Author Response

Minor

  1. I suggest you add a picture showing the chemical structure of all mentioned (apparently) highly potent and well-tolerated RET-altered inhibitors (e.g., selpercatinib, pralsetinib, TPX-0046, zetletinib). I suggest adding a picture showing the chemical structure of all mentioned (apparently) highly potent RET-altered inhibitors. This information is important for (organic) chemists and biochemists.

Response: Thanks for your suggesting. We added selpercatinib, pralsetinib, TPX-0046, zetletinib chemical structure in figure 2. (line 227-229)

  1. Two sentences are inappropriate beginning parts of the text... please correct them. The first is on line 208, and the second starts on line 233.

Response: Thanks for your suggesting, we change inappropriate sentences from line 208 to “MKIs has lots of disadvantages such as adverse events and drug resistance” and line 233 to “The pivotal clinical trial LIBRETTO-001 (NCT03157128)”

  1. Please change the title of Figure 1. It shows a schematic representation of the structure of RET (and activation mechanism). The first letter must be capitalised in this figure and in the title of Table 1. Please use Ca2+ (number and plus sign in superscript), not Ca++.

Response: Thanks for your attention. The content of title of Figure 1 has been adjusted in accordance with the recommendations.

  1. Almost the entire text lacks a space between the last word in the sentence, and the parentheses for the reference cited (lines 31, 34, 37, etc.)

Response: We added all the space between the last word in the sentence and the parentheses for the reference cited. 

5. Lines 290 and 291: according to the rest of the text, write n=1, not n = 1 (so no spaces)! Next, move the period/dot to the end of the sentence (lines 76 and 294, the end of the fourth paragraph of section 3.2.2. and section 4).

Response: Thanks for your attention. The content has been adjusted in accordance with the recommendations (line 291-292).

6. The name of the drug is incorrectly stated (typing error) in the sentence in the second paragraph of section 3.2.2. (Furthermore, ...).

Response: All the incorrectly words (pralsetinib) has been correct, and we wish you have a nice day.

Reviewer 2 Report

The review article manuscript of Gou et al. deals with an interesting topic of RET inhibitors. I found that the manuscript is lousy and needs a major revision before publication. 

1.      Please add the following references in the relevant part of the article described in the attached file.

a.      https://www.frontiersin.org/articles/10.3389/fphys.2019.00216/full

b.     https://cshperspectives.cshlp.org/content/5/2/a009134.short

2.     The * and # symbols are either not used properly or not explained well.

3.     RET also stands for Rearranged during transfection and do mention this in beginning of the article. I saw this as the most commonly described expansion.

4.     Check line 8 and remove corresponding to

5.     On line 6, mention which figure you are talking about.

6.     Improve the quality of figure 1.

7.      It isn't easy to follow the article because many technical words are not expanded when they are used for the first time in the article.

8.      The table also needs correction, the name of drugs are either misplaced or not mentioned. This confuses me in understanding the data mentioned in the table.

9.     The references need to be written in a proper format, please have a thorough look and fix the issues, e.g., in some references journal name is abbreviated, while in others it is not. Most of the time, references are not consistent – please have a careful look.

Author Response

reviewer 2

major

  1. Please add the following references in the relevant part of the article described in the attached file.

  1. https://www.frontiersin.org/articles/10.3389/fphys.2019.00216/full

  1. https://cshperspectives.cshlp.org/content/5/2/a009134.short

Response:

Thanks for your suggestion. We added these references to line 59 (a: No. 24) and line 79 (b: No. 31). These references can make this article more rigorous.

  1. The * and # symbols are either not used properly or not explained well.

Response: We correct the symbols and exlplain to you. “*” symbolized co-author, it means they contributed equally work, and Qitao Gou, Xiaochuan Gan, Qiheng Gou, contributed equally in this article. “#” was symbolized corresponding author, Qiheng Gou and Tao Zhang are corresponding author.

  1. RET also stands for Rearranged during transfection and do mention this in beginning of the article. I saw this as the most commonly described expansion.

Response: In the abstract, we added the content to the article (line 12-13).

  1. Check line 8 and remove corresponding to

Response: We remove “corresponding to” to the article, and thanks for your attention. (line 7)

  1. On line 6, mention which figure you are talking about.

Response: The content has been adjusted in accordance with the recommendations (line 80).

  1. Improve the quality of figure 1.

Response: Thanks for your attention. Because of the format was changed by editor, the quality of the figure is not good to read. We adjusted the figure 1 size accordance with the recommendations.

  1. It isn't easy to follow the article because many technical words are not expanded when they are used for the first time in the article.

Response: I sincerely apologize for my negligence. The ORR, PFS, CR, PR, OS was expanded when they are used for the first time in the article. (line 129,130, 150, 151, 153-154 respectively)

  1. The table also needs correction, the name of drugs are either misplaced or not mentioned. This confuses me in understanding the data mentioned in the table.

Response: In this table, we apologized for negligence in our work. We deleted not mentioned drug and clinical trials. Besides, because of editor change the format, some location is not easy to understand the table. We corrected the location of this drug name, and make them easier to understand in this table.

  1. The references need to be written in a proper format, please have a thorough look and fix the issues, e.g., in some references journal name is abbreviated, while in others it is not. Most of the time, references are not consistent – please have a careful look.

Response: Thanks for your attention. We corrected the right reference format and fix this issue.

Round 2

Reviewer 2 Report

I am glad to see changes made in the manuscript. I believe that this manuscript will be useful for the reader of Molecules. I still see some issues with the references, please have a careful look e.g.,

1. 65, 70, 77, 80, 81, 82, 85, 86 etc have full name of journals in references

2. Please add doi in all reference e.g., reference 9, 85, 86 etc